# Calf Circumference as an Indicator for Cystatin C Testing in Hospitalized Elderly Male Patients for Detecting Hidden Renal Impairment

**DOI:** 10.3390/jcm12216899

**Published:** 2023-11-02

**Authors:** Sunghwan Lee, Seul Lee, Sunhee Jo, Kyung-Ah Kim, Yu Jin Yang, Jung Joo Lee, Eunsung Kim, Yongjung Park, Taeyoung Kyong, Jeong-Ho Kim

**Affiliations:** 1Department of Laboratory Medicine, Yongin Severance Hospital, Yonsei University College of Medicine, 363 Dongbaekjukjeon-daero, Giheung-gu, Yongin-si 16995, Gyeonggi-do, Republic of Korea; gnsguzzy@yuhs.ac (S.L.); kyunga4566@yuhs.ac (K.-A.K.); 2Department of Hospital Medicine, Yongin Severance Hospital, Yonsei University College of Medicine, 363 Dongbaekjukjeon-daero, Giheung-gu, Yongin-si 16995, Gyeonggi-do, Republic of Korea; seul21104@yuhs.ac (S.L.); xxshjo@yuhs.ac (S.J.); 3Department of Nutrition, Yongin Severance Hospital, Yonsei University College of Medicine, 363 Dongbaekjukjeon-daero, Giheung-gu, Yongin-si 16995, Gyeonggi-do, Republic of Korea; yyujin90@yuhs.ac (Y.J.Y.); newtree@yuhs.ac (J.J.L.); 4Department of Inpatient Nursing, Yongin Severance Hospital, Yonsei University College of Medicine, 363 Dongbaekjukjeon-daero, Giheung-gu, Yongin-si 16995, Gyeonggi-do, Republic of Korea; happyes9@yuhs.ac; 5Department of Laboratory Medicine, Gangnam Severance Hospital, Yonsei University College of Medicine, 211, Eonju-ro, Gangnam-gu, Seoul 06273, Republic of Korea; ypark119@yuhs.ac

**Keywords:** calf circumference, sarcopenia, eGFR (estimated glomerular filtration rate), cystatin C, creatinine

## Abstract

Serum creatinine is used to measure the estimated glomerular filtration rate (eGFR); however, it is influenced by muscle mass and may therefore overestimate renal function in patients with sarcopenia. We examined calf circumference (CC) as a convenient muscle mass evaluation tool that can potentially indicate the need to test for cystatin C instead of creatinine in elderly inpatients. We retrospectively reviewed the electronic health record of 271 inpatients aged 65 or over. CC was determined by measuring the thickest part of the nondominant calf. eGFRcys and eGFRcr were calculated using cystatin C and creatinine levels, respectively. We evaluated optimum CC cutoff values using the eGFRcys/eGFRcr ratio for detecting hidden renal impairment (HRI, defined as eGFRcr ≥ 60 mL/min/1.73 m^2^ but eGFRcys < 60 mL/min/1.73 m^2^). CC showed a significant positive correlation with the eGFRcys/eGFRcr ratio in both sexes. The areas under the receiver operating characteristic curve were 0.725 and 0.681 for males and females, respectively. CC cutoffs with a sensitivity or specificity of 90% or 95% might be used to detect HRI in males. In conclusion, utilizing the optimum cutoff, CC could be a cost-effective screening tool for detecting HRI in elderly male patients using cystatin C as an add-on test.

## 1. Introduction

Evaluation of renal function using the estimated glomerular filtration rate (eGFR) is routinely performed during patient management. Serum creatinine is widely used as an endogenous marker to calculate eGFR because creatinine assays are standardized and available in most clinical laboratories [1]. Although eGFR calculation based on serum creatinine is convenient and cost-effective, the serum creatinine level is influenced by several non-GFR factors, including age, sex, drugs, certain chronic conditions, diet, and muscle mass, among which lean muscle mass is a major contributor [2]. Low muscle mass leads to lower endogenous creatinine production, which may lead to an overestimation of renal function because eGFR equations for creatinine do not consider muscle mass. According to a previous study by Andrade et al., approximately 30% of hospitalized patients experience protein malnutrition, and muscle mass loss in the intensive care unit varies from 17% to 30% during the first 10 days of admission [3]. Therefore, when evaluating elderly inpatients with or at risk of developing sarcopenia, an endogenous marker other than creatinine may be more appropriate for calculating eGFR [2]. Among the many markers, serum cystatin C has been suggested as an alternative to creatinine in patients with presumed sarcopenia because it is not influenced by muscle mass [2,4]. However, the exact clinical indications for using serum cystatin C instead of serum creatinine are unclear. Yim et al. [5], in a previous study, investigated skeletal muscle index (SMI) as a surrogate marker for sarcopenia and proposed SMI values of 7.3 kg/m^2^ for males and 5.7 kg/m^2^ for females as significant cutoffs for indication of cystatin C test. Although this study provided a numerical cutoff, calculating SMI itself requires bioelectrical impedance analysis (BIA), which limits its broad bedside applications. Other methods for assessing muscle mass are magnetic resonance imaging, computerized tomography, and dual-energy X-ray absorptiometry (DXA) [6] However, they are also unsuitable for broad usage because of their cost and/or potential radiation exposure. To simplify decision making, several studies have suggested calf circumference (CC) measurement as a quick surrogate marker for determining sarcopenia and, consequently, as an indication for cystatin C measurement [5,6,7]. In this study, we aimed to investigate the validity of CC as an indicator for cystatin C measurement and provide specific cutoff values for patients over the age of 64.

## 2. Materials and Methods

### 2.1. Patients

Electronic health records (EHRs) of inpatients hospitalized at Yongin Severance Hospital (Yongin, Republic of Korea) between 1 March 2020 and 30 June 2023 were reviewed. We utilized a clinical research data warehouse known as the Severance Clinical Research Analysis Portal. This platform enabled us to extract specific clinical data from pseudonymized patient records, providing us with the following information: age at the time of CC measurement, sex, serum creatinine, serum cystatin C, and CCs of both legs. Our study focused on elderly patients aged 65 and older, for whom these parameters were measured upon admission. CC is a parameter in the EHR that is used to assess muscle mass or edema. Due to limited manpower, CC measurements were not available for all inpatients. To ensure concordance and accuracy, only patients who had undergone blood tests within a week of CC measurement and had also been tested for serum creatinine and cystatin C simultaneously were chosen for final data analysis.

Patient exclusion criteria considered were as follows:(1)Body mass index (BMI) ≥ 30 kg/m^2^;(2)Patients wearing compression stockings;(3)Patients unable to assume proper leg position due to disability or recent surgery;(4)Amputees with only one leg.

### 2.2. Calf Circumference (CC) Measurement

CC was measured in either a sitting or supine position. In the sitting position in a chair or wheelchair, patients were instructed to maintain a neutral posture with 90-degree flexion of the knees and ankles, with the sole resting on the floor or a footrest. If the patient could not sit comfortably, they were instructed to lie supine with their knees flexed at a 90-degree angle, and the sole of the footrest on a bed or sandbag. A nonelastic tape ruler was used to measure CC. The tape was passed around each calf without compressing the skin. It was gently slid up and down to identify the point on the calf with the largest circumference. Measurements were taken at that point on both calves [8]. After each measurement, the measuring tape ruler was disinfected with isopropyl alcohol to ensure hygiene before using it on the next patient. For more information or visual aids regarding the measuring process, Andrade et al. provided additional details [3].

After measurement, the larger calf circumference measurement was designated as the ‘dominant calf’, and the smaller value was designated as the ‘non-dominant calf’.

### 2.3. Estimation of Renal Function 

Serum creatinine and cystatin C levels were measured in the same blood samples from each patient.

Serum creatinine levels were measured using the enzymatic method (Roche Creatinine Plus ver.2 assay), which was standardized against the isotope dilution mass spectrometry method.

Serum cystatin C levels were measured using an immunoturbidimetric method (Tina-quant Cystatin C Gen. 2, Basel, Switzerland, Roche), which is standardized and traceable against the ERM-DA471/IFCC reference material.

Both tests were performed using Roche cobas 8000 c 702 (Roche Diagnostics, Basel, Switzerland).

eGFR based on creatinine (eGFRcr) was calculated using the Chronic Kidney Disease Epidemiology Collaboration (CKD-EPI) 2009 equation [9]. 

We employed the CKD-EPI 2009 creatinine equation, which has previously been validated as suitable for the Korean population, as demonstrated by two studies that compared it with measured GFR using ^51^Cr-EDTA [10,11]. Specifically, we used the CKD-EPI (2009) non-Black equation, as all the participants in our study were of Korean ethnicity.

eGFR based on cystatin C (eGFRcys) was calculated using the CKD-EPI cystatin C 2012 [12].

As one of the criteria for a chronic kidney disorder is defined by eGFR ≤ 60 mL/min/1.73 m^2^, for this study we defined hidden renal impairment (HRI) as the condition of having eGFRcr ≥ 60 mL/min/1.73 m^2^ while eGFRcys < 60 mL/min/1.73 m^2^ concurrently. This definition is meant to identify patients whose creatinine levels are unreliable as markers of renal function owing to reduced muscle mass.

### 2.4. Statistical Analysis

The Mann–Whitney U test was used to compare continuous variables between the two groups. The Spearman correlation coefficient (*r*_s_) was used to analyze the correlation between two parameters. Receiver operating characteristic (ROC) curves for CC and HRI were constructed, and the cutoff values of CC for the detection of HRI were determined. Significant differences in area under the receiver operating characteristic curve (AUROC) were compared by random chance. Statistical analyses were performed with Analyse-it version 5.92 for Microsoft Excel 2021 (Analyse-it Software Ltd., Leeds, UK).

## 3. Results

### 3.1. Patient Demographics and Baseline Measurements

In total, 271 patients (135 men and 136 women) were included in this study. Pertinent measurement values, including nondominant calf circumference, serum creatinine and cystatin C levels, and various eGFR calculations, are summarized in Table 1.

To preclude the possibility of obesity confounding the interpretation of cystatin C levels, only patients with a BMI < 30 kg/m^2^ were included. The BMI comparison of both sexes showed no statistically significant differences, as shown in Table 1.

### 3.2. Relationship between eGFRcr and eGFRcys

eGFRcr and eGFRcys were compared in sarcopenia screening positive vs. negative groups, as well as in male vs. female groups, to confirm that both values showed a positive correlation in all groups of patients (Figure 1). We used the Asian Working Group for Sarcopenia (AWGS) 2019 consensus definition (CC < 34 cm for males and <33 cm for females) to screen for sarcopenia [12]. The correlation was measured at 0.853 (95% confidence interval (CI) 0.774–0.854) using Spearman’s rank correlation coefficient (*r*_s_), with a *p*-value < 0.0001.

### 3.3. Relationship between Calf Circumference and eGFR Ratio

The eGFRcys/eGFRcr ratio was used to assess the disparity between the two renal function indicators, and its correlation with CC is shown in Figure 2. As expected, there was a significant positive correlation between the eGFRcys/eGFRcr ratio and nondominant CC, suggesting that eGFRcr may be overestimated in patients with sarcopenia. A significant positive correlation was observed, with *r*_s_ = 0.434 (95% CI: 0.211–0.621; *p* = 0.0002) for males, and with *r*_s_ = 0.509 (95% CI, 0.310–0.666; *p* < 0.0001) for females.

### 3.4. The Proportion of Hidden Renal Impairment According to eGFRcr

To further highlight the prevalence of HRI cases, we categorized patients into renal impaired and nonimpaired groups, similar to the definition of CKD (eGFR < 60 mL/min/1.73 m^2^) (Figure 3). Among the patients classified as impaired based on eGFRcr, 94.85% were also identified as renally impaired based on eGFRcys. However, in the group classified as not impaired by eGFRcr, only 60.28% were not impaired by eGFRcys, whereas 39.72% were renally impaired (Figure 3). These findings highlight the presence of significant cases of HRI in the study group. 

### 3.5. ROC Curves

To determine the optimal CC threshold for detecting HRI, we calculated the ROC curves for males and females (Figure 4). In the male group, the AUROC was 0.725 (95% CI: 0.594–0.855; *p*—0.0007). In the female group, the AUROC was 0.681 (95% CI: 0.556 to 0.805; *p*—0.0045). The decision threshold graphs show the cutoffs for 95% sensitivity and the ‘optimum threshold’ as determined by Analyze-it software. For more comprehensive data, we calculated the CC cutoff corresponding to 90% or 95% sensitivity or specificity for each parameter according to sex.

We also calculated and compared the AUROC values for the dominant, nondominant, and average CC to determine the best CC measure for detecting HRI (Table 2). The median coefficient of the left and right CC was 1.07% (25th percentile, 0.43%; 75th percentile, 1.89%; 90th percentile, 3.24%; maximum, 9.69%). For males, the AUROC for the average and nondominant CC were similar. However, among females, the nondominant CC appeared slightly better than the others, although with no significant difference. Therefore, the authors opted for nondominant CC for detecting HRI to maintain consistency in our study. 

### 3.6. Calf Circumference Cutoff Values

We calculated specific nondominant CC cutoff values for detecting HRI, as shown in Table 3 and Table 4. For specificities of 90% and 95%, CC cutoffs were 29.6 and, 29.1 cm for males, and 26.0 and, 23.2 cm for females. For sensitivities of 90% and 95%, CC cutoffs were 35.5 and 37 cm for males, and 33.6 and 34.1 cm for females. In addition, we listed data for CC of 34 cm (male) and 33 cm (female), which correspond to the sarcopenia screening criteria in the AWGS 2019 Consensus Update [13].

## 4. Discussion

Serum creatinine and associated eGFRcr levels are frequently used to evaluate renal function in inpatients. However, it is well known that serum creatinine levels are highly dependent on total muscle mass [2,14]. Because eGFRcr equations do not consider muscle mass, they tend to overestimate renal function in patients with sarcopenia. Elderly patients are particularly affected because aging is an important factor in sarcopenia. It has been suggested that cystatin C can be used as an alternative endogenous marker to evaluate renal function in patients having or likely to have sarcopenia because it is unaffected by muscle mass [2]. However, the current diagnostic criteria for sarcopenia require multifaceted evaluations, including the assessment of muscle strength, physical performance, and appendicular skeletal muscle mass using either DXA or BIA. These tests are expensive, time-consuming, and sometimes challenging to perform, particularly for uncooperative patients. Therefore, attempts have been made to establish convenient sarcopenia screening tools. Our study builds on previous investigations suggesting that CC is correlated with total muscle mass [7,15,16]. and it has also been validated in large population studies such as NHANES [6]. An international survey indicated that CC is the most popular method for assessing muscle mass followed by DXA, skinfold thickness, and BIA [17]. Many studies have explored the appropriate cutoff value of CC for sarcopenia screening [6,7,15]. However, there is limited research available for CC establishing cutoff value specifically for detecting HRI [5]. However, that study employed CC measurement at the midpoint between the lateral epicondyle of the distal femur and the prominent point of the fibula’s lateral malleolus bone, which was not a widely accepted method for measuring CC [5,18]. In this current study, we employed a more universally accepted method, measuring CC at the thickest point of both calves in this study [5,7,8]. Since HRI is expected to be directly related to sarcopenia by principle, we attempted to use CC as a screening test and established cutoff values for ordering a cystatin C test in elderly patients.

For this study, we reviewed the electronic medical records of 271 patients over the age of 64. The basic demographic data presented in Table 1 indicate that males exhibited significantly larger nondominant CC compared to females. Of note, it also shows that while serum creatinine was higher in males, serum cystatin C did not show a statistically significant difference between the sexes, corroborating the assumption that it is not affected by total muscle mass.

The eGFRcr and eGFRcys scattergrams in Figure 2 show a high correlation in the study population regardless of sex or sarcopenia status, indicating that both measures reflect renal function. The higher prevalence of sarcopenia in quadrant 4 is worth noting. These cases in quadrant 4 are also classified as HRI, characterized by an eGFRcr ≥ 60 mL/min/1.73 m^2^, with a concurrent eGFRcys < 60 mL/min/1.73 m^2^. A closer look at the eGFRcys/eGFRcr ratio versus the nondominant CC revealed a trend of divergence as muscle mass either increased or decreased. This relationship corroborates our assumption that muscle mass affects eGFRcr.

CC has been proposed as an anthropometric estimation of muscle mass when no other method is available for the diagnosis of malnutrition or sarcopenia and is a simple, cost-effective screening test for sarcopenia or malnutrition [19].

The positive or negative likelihood ratio indicating a cystatin C test using CC is not as high in females as in males (Table 3). However, the eGFRcys/eGFRcr ratio was correlated with CC, as shown in Figure 2, similar to the findings of Yoshida et al. [20]. Additionally, the AUROC values are similar to those reported by Kusunoki et al. [21]. 

We propose incorporating serum cystatin C testing in addition to serum creatinine measurement if the CC of the nondominant side is less than 29.6 cm in males (see Table 3). For males with a CC exceeding 35.5 cm (Table 3), the inclusion of the cystatin C add-on test may not be necessary. However, the utility of CC as a surrogate marker for guiding serum cystatin C testing in females is less straightforward.

In the present study, the use of CC to detect HRI was less robust in women than in men. Although the AUROC for the female group showed statistical significance, CC was not as indicative of HRI in females as in males. There have been studies where eGFRcr was found to be affected by muscle mass in elderly males, but less significantly in elderly females [22]. This may explain why our results were not as robust as expected. Another possibility is that CC may not strongly represent total muscle mass in elderly females, as fat and muscle distribution may be affected differently based on sex and age. For example, research has shown that elderly women tend to have more subcutaneous fat than elderly men [23]. We postulate two potential reasons for the lower predictive values of CC in elderly women compared to elderly men. (1) It is known that muscle mass and, accordingly, serum creatinine levels decrease to a greater extent in elderly men compared to that in elderly women. Consequently, the eGFRcys/eGFRcr ratio might be less affected by sarcopenia in elderly women than in elderly men. (2) Furthermore, due to the higher prevalence of sarcopenic obesity compared to men, CC may not be a reliable indicator of total muscle mass among women [24]. Unfortunately, we could not evaluate sarcopenic obesity in this study, and additional research is necessary to reach a definitive conclusion. One must also keep in mind that CC criteria in the ‘AWGS 2019 Consensus’ is meant to be a screening tool for sarcopenia, not as an accurate surrogate for total muscle mass [12]. The European consensus on the definition and diagnosis of sarcopenia guidelines also acknowledges the limitation of anthropometric measures in estimating muscle mass and suggests their use in cases where formal muscle mass evaluation is difficult [25]. We suspect that these inherent limitations of CC contributed to the varying levels of robustness of our results.

Many anthropometric measures have been studied as surrogate markers for total muscle mass, including weight, height, arm circumference, arm muscle circumference, triceps skinfold thickness, adductor pollicis muscle thickness, and CC. We chose to use CC in our study because it is thought to be a more sensitive measure for elderly patients [5,24].

However, although CC itself has been well investigated, to the best of our knowledge, there has been no consensus on which CC value (average vs. dominant vs. nondominant) best reflects total muscle mass. Studies have varied in the choice of CC values. To determine which CC was best suited for detecting HRI, we calculated and compared the AUC values of three ROC curves (average, dominant, and nondominant CC). In males, the AUC of the average and nondominant CCs were very closely matched, but the nondominant CC showed apparent superiority in females. Therefore, we recommend that the nondominant CC be used to detect potential HRI.

When a major discrepancy between eGFRcr and eGFRcys is detected, physicians would need to investigate the possible causes and determine which equation is more credible given the patient’s clinical situation. For instance, in an elderly patient with sarcopenia and no evidence of thyroid disease, obesity, or steroid use, eGFRcys is more likely to represent the true GFR. In a study by Ebert et al., the authors suggested that a thorough investigation is required if the eGFR discrepancy exceeds 40% and proposed that if neither eGFR value can be trusted, a measured GFR (mGFR) test should be carried out [2].

The limitations of this study include the following. (1) The gold standard for directly measured GFR [1,26] using exogenous markers, such as inulin, iothalamate, iohexol, or ^51^Cr-EDTA, was not employed in this study. While we used eGFRcys ≤ 60 mL/min/1.73 m^2^ as the criteria for HRI, it is important to note that cystatin C itself is an endogenous marker, which can only provide an estimate of the true GFR. Additionally, we did not control for factors that could influence the accuracy of serum cystatin C levels. Several factors such as obesity, chronic inflammation, thyroid disease, and steroid usage affect cystatin C levels. Although we excluded patients with BMI ≥ 30 kg/m^2^, these factors may have introduced variability into our results. Despite this limitation, cystatin C is recommended as a confirmatory test owing to its routine availability from a single blood sample compared with the direct measurement of GFR [1]. (2) Although well-trained nurses and medical technologists measured CC and we occasionally checked the reproducibility of CC measurement by two raters for the same patients, there may still be some uncertainties related to inter-rater variability. However, we are confident that CC measurement variability was minimal, as we measured both the right and left CC and corrected for any measurement errors. (3) The retrospective cross-sectional nature of our study implies the presence of inherent bias and the challenge of demonstrating a difference in patient outcomes based on the eGFRcr/eGFRcys ratio. (4) Our findings may not be generalizable to other populations because this study was relatively small and involved only 271 participants from a single institution in Korea. (5) Furthermore, as the study subjects were not randomly selected from inpatients aged 65 and older with available manpower to measure CC upon admission, there may be a selection bias. Consequently, this study could be considered a pilot study. (6) We could not entirely eliminate the possibility of CC measurement being affected by the presence of generalized edema and variation in the hydration status of inpatients. Future large-scale studies with stringent exclusion criteria and direct measurement of the GFR may be necessary to refine the CC cutoff calculations to detect HRI.

## 5. Conclusions

The disparity between eGFRcr and eGFRcys was more pronounced among elderly male patients with a lower CC. The CC cutoff values proposed in our study can serve as a practical screening tool to determine whether a patient requires a serum cystatin C test for a more accurate evaluation of renal function.

## Figures and Tables

**Figure 1 jcm-12-06899-f001:**
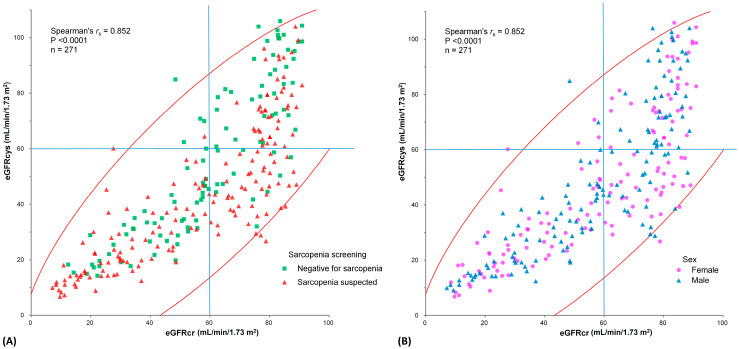
Scattergram between estimated glomerular filtration rates determined by cystatin C (eGFRcys) and creatinine (eGFRcr) of study participants. There was a positive correlation between eGFRcr and eGFRcys (n = 271, *r*_s_ = 0.853; 95% confidence intervals, 0.774 to 0.854, *p* < 0.0001). (**A**) We symbolized each data with different colors and shapes as follows: triangular shape with cyan color for sarcopenia suspected case, rectangular shape in green color for normal case, with nondominant calf circumference according to Asian Working Group for Sarcopenia (AWGS) 2019 guideline. An observation worth noting is that a higher prevalence of sarcopenic cases is in quadrant 4. These cases are also classified as HRI, characterized by having eGFRcr ≥ 60 mL/min/1.73 m^2^, while concurrently eGFRcys < 60 mL/min/1.73 m^2^. (**B**) We symbolized each data with different colors and shapes as follows: triangular shape with cyan color for male case, rectangular shape in green color for female case, with nondominant calf circumference according to AWGS 2019 guidelines. The red elliptical lines represent the density ellipse of 95% of the subjects.

**Figure 2 jcm-12-06899-f002:**
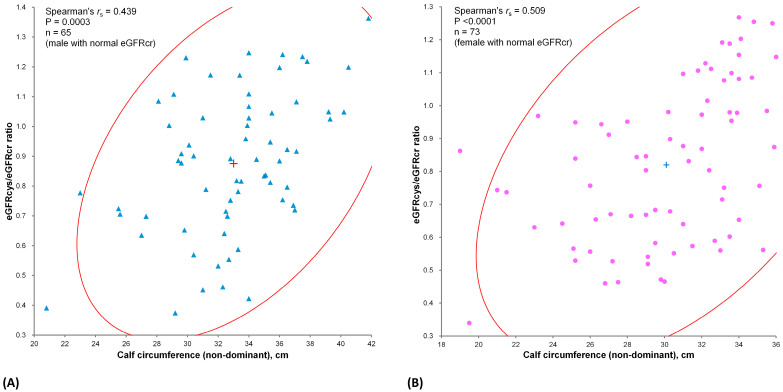
Correlation between nondominant calf circumference and estimated glomerular filtration rates determined by cystatin C (eGFRcys) to eGFR by creatinine (eGFRcr) ratio: (**A**) male group; (**B**) female group. Blue triangles represent subjects in the correlation of eGFRcys/eGFRcr ratio to calf circumference in the male group (**A**), while pink circles represent subjects in the female group. The cross symbols of both graph represent the center of correlation. The red elliptical lines represent the density ellipse of 95% of the subjects.

**Figure 3 jcm-12-06899-f003:**
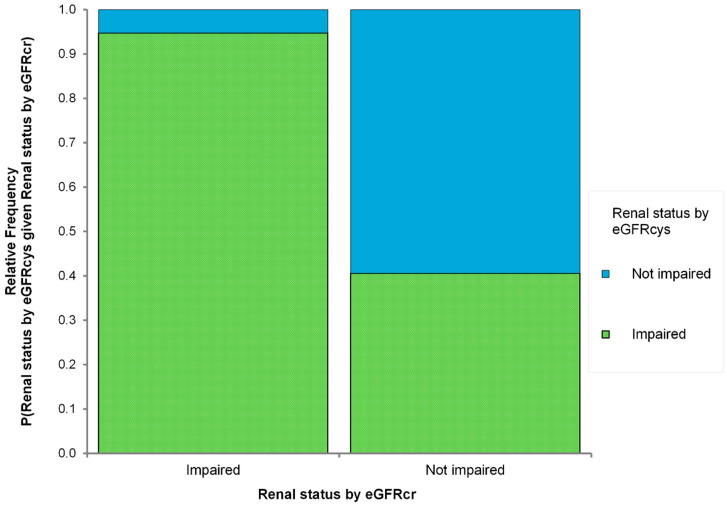
Proportion of either “Impaired” or “Not impaired” cases according to estimated glomerular filtration rate determined by cystatin C (eGFRcys) or creatinine (eGFRcr).

**Figure 4 jcm-12-06899-f004:**
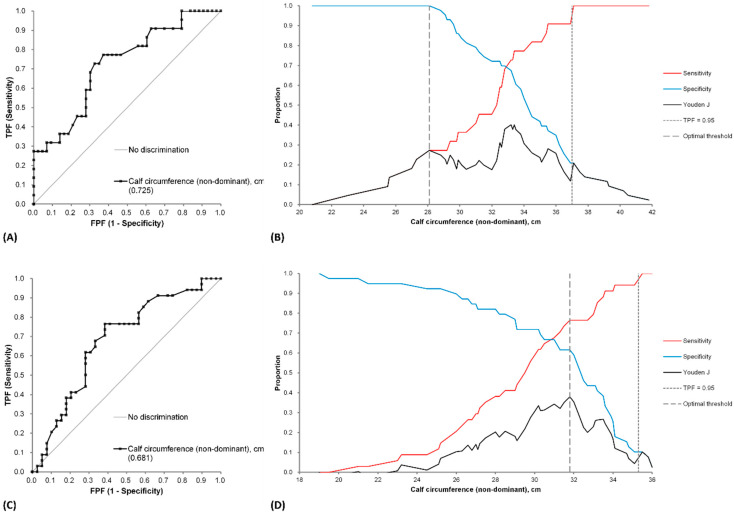
Receiver operating characteristic (ROC) curves and decision threshold curves were used to determine the cutoff value of calf circumference for the detection of hidden renal impairment with 95% sensitivity for males and females: (**A**) ROC curve for males; (**B**) decision threshold curve for males; (**C**) ROC curve for females; (**D**) decision threshold curve for females.

**Table 1 jcm-12-06899-t001:** Patient characteristics.

	Male	Female	*p*-Value *
Number	135	136	
Age	78.0 (71.0–83.0)	79.0 (75.0–85.0)	**0.0085**
Calf circumference, nondominant	33.5 (31.0–35.92)	31.0 (28.0–33.5)	**<0.0001**
eGFRcr ^†^	59.1 (38.2–78.0)	62.3 (34.6–80.1)	0.8581
eGFRcys ^‡^	44.0 (28.3–68.1)	44.5 (26.15–66.5)	0.7181
Creatinine, mg/dL	1.14 (0.90–1.65)	0.88 (0.65–1.44)	**0.0031**
Cystatin C, mg/L	1.44 (1.04–2.03)	1.35 (0.987–2.013)	0.514
BMI	23.53 (21.03–25.57)	23.23 (21.25–25.79)	0.9337

* Continuous variables are presented as median and interquartile ranges (first and third quartiles) and were compared between the two groups using the Mann–Whitney U test. Bold formatting was used to indicate significant differences (*p* value < 0.05). ^†^ Estimated by the Chronic Kidney Disease Epidemiology Collaboration (CKD-EPI) creatinine equation, 2009 version. ^‡^ Estimated by the CKD-EPI cystatin C equation, 2012 version. eGFRcr, eGFR based on creatinine; eGFRcys, eGFR based on cystatin C; BMI, body mass index. Of note, the male and female groups showed significant differences between the nondominant CC and serum creatinine values, underlining the difference in baseline muscle mass between the two groups, even among the elderly population.

**Table 2 jcm-12-06899-t002:** Comparison of AUROC using different calf circumference parameters for hidden renal impairment detection.

Parameter	Male		Female	
	AUROC	*p*-Value	AUROC	*p*-Value
Average	0.726 (0.595 to 0.856)	0.0007	0.675 (0.550 to 0.801)	0.0062
Dominant	0.720 (0.588 to 0.853)	0.0011	0.666 (0.539 to 0.792)	0.0105
Nondominant	0.725 (0.594 to 0.855)	0.0007	0.681 (0.556 to 0. 805)	0.0045

AUROC, area under a receiver operating characteristic curve (95% confidence intervals); average, average value of left and right calf circumference in cm; dominant, the larger value of left and right calf circumference in cm; nondominant, the smaller value of left and right calf circumference in cm. We did not observe any statistical differences among AUROCs of average, dominant, and nondominant values of calf circumference when detecting hidden renal impairment in either the male or female group using a significance level of *p* = 0.05.

**Table 3 jcm-12-06899-t003:** Calf circumference cutoff for various sensitivity/specificities for males.

Calf, Circumference (cm), Nondominant	Label	Sensitivity	Specificity	Likelihood Ratio (+)	Likelihood Ratio (−)	Youden’s Index
29.1	Specificity~95%	0.273	0.957	6.27	0.76	0.229
29.6	Specificity~90%	0.318	0.913	3.66	0.75	0.231
34.0	AWGS (2019) *	0.773	0.543	1.69	0.42	0.316
35.5	Sensitivity~90%	0.909	0.391	1.49	0.23	0.300
37.0	Sensitivity~95%	0.955	0.217	1.22	0.21	0.172

* AWGS (2019), sarcopenia screening criteria in Asian Working Group for Sarcopenia: 2019 Consensus Update [13].

**Table 4 jcm-12-06899-t004:** Calf circumference cutoff for various sensitivities and specificities for females.

Calf Circumference (cm), Nondominant	Label	Sensitivity	Specificity	Likelihood Ratio (+)	Likelihood Ratio (−)	Youden’s Index
23.2	Specificity~95%	0.088	0.949	1.72	0.96	0.037
26.0	Specificity~90%	0.206	0.897	2.01	0.88	0.103
33.0	AWGS (2019) *	0.794	0.436	1.41	0.47	0.230
33.6	Sensitivity~90%	0.912	0.333	1.37	0.26	0.245
34.1	Sensitivity~95%	0.941	0.179	1.15	0.33	0.121

* AWGS (2019), please see the footer of Table 3.

## Data Availability

The datasets used and analyzed in this study are available from the corresponding authors upon an appropriate request, subject to written approval by the Yonsei University Health System.

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
