# Peer review of "Calf Circumference as an Indicator for Cystatin C Testing in Hospitalized Elderly Male Patients for Detecting Hidden Renal Impairment"

_jcm, 2023, doi:10.3390/jcm12216899_

Round 1

Reviewer 1 Report

Comments and Suggestions for Authors

Dear authors. The work is interesting in its concept and it seems to me that it can be useful for the practical activities of doctors. The scientific part and methodology of the study does not stand up to criticism. First, it seems to me that the authors must understand for themselves what they want to present. If the idea is like a scientific article, then it is necessary to make a comparison with the impedance as a minimum. And also with the general hydration of the patient, since if kidney function is limited, there will be fluid retention. If the authors are only focused on the practical aspect for the doctor without a scientific basis, I am inclined to think that it is necessary to carefully describe the methodology: where and how the measurement is taken (give a picture), explain why the non-dominant calf of the CC is chosen, and not the average between the largest and smallest, give an explanation of how to take into account the presence of edema and general hydration. It may be more correct to measure the CC on the thigh, where tissue swelling has less of an effect on the result. At a minimum, it should be noted that the method is only suitable for men and without the presence of edema. You can suggest weighing the patient on an impedance scale, and in the absence of edema, you can use the proposed technique. But impedance scales show muscle mass, which may suggest sarcopenia. Impedance scales are not expensive. In my point of view, the article has to be improved

Reviewer 2 Report

Comments and Suggestions for Authors

The authors have attempted to investigate the validity of calf circumference (CC) as an indicator for cystatin C measurement and provide specific cutoff values for patients over the age of 64.

The study is methodologically well performed.

There are a few of points which may be considered for further improvement.

271 patients were considered eligible. Reasons for ineligibility have not been mentioned clearly with numbers.

The number of patients included in this study could be higher. Selection bias is present, but it can be tolerated considering that it is a pilot study.

Please provide reference justifying However, it is well known that serum creatinine levels are highly dependent on total muscle mass” Line 255

Provide references justifying Many studies have explored the appropriate cutoff value of CC for sarcopenia screening; however, there is no established cutoff value for detecting HRI.” Line 267

Should be stated clearly in the limitation section: it is inherently limited by biases related to its retrospective cross-sectional design.

The findings of this study could be of importance for clinical practice.

Round 2

Reviewer 1 Report

Comments and Suggestions for Authors

Dear authors. I am grateful to you for your detailed, comprehensive explanations and clarifications. A more extend references is always welcome. And I didn’t see the photographs with the method in the article that you presented in your letter to me.
